# A Systematic Review and Meta-Analysis of the Effect of Caloric Restriction on Skeletal Muscle Mass in Individuals with, and without, Type 2 Diabetes

**DOI:** 10.3390/nu16193328

**Published:** 2024-09-30

**Authors:** Oluwaseun Anyiam, Rushdina Sofia Abdul Rashid, Aniqah Bhatti, Saif Khan-Madni, Olakunmi Ogunyemi, Arash Ardavani, Iskandar Idris

**Affiliations:** 1MRC/ARUK Centre for Musculoskeletal Ageing Research and National Institute for Health Research (NIHR), Nottingham Biomedical Research Centre (BRC), School of Medicine, University of Nottingham, Royal Derby Hospital Centre, Derby DE22 3DT, UK; 2Department of Endocrinology and Diabetes, University Hospitals of Derby and Burton NHS Foundation Trust, Derby DE22 3NE, UK; 3Department of Endocrinology and Diabetes, United Lincolnshire Hospitals NHS Trust, Lincoln LN2 5QY, UK; 4Nottingham University Hospitals NHS Trust, Nottingham NG7 2UH, UK; 5School of Medicine, University of Nottingham, Nottingham NG7 2RD, UK; 6Department of Acute Medicine, University Hospitals of Derby and Burton NHS Foundation Trust, Derby DE22 3NE, UK; 7Nuffield Department of Population Health, University of Oxford, Oxford OX1 2JD, UK

**Keywords:** type 2 diabetes, obesity, caloric restriction, very-low-calorie diet, low-calorie diet, body composition, muscle mass

## Abstract

Background. Severe caloric restriction interventions (such as very-low-calorie diets) are effective for inducing significant weight loss and remission of type 2 diabetes (T2DM). However, suggestions of associated significant muscle mass (MM) loss create apprehension regarding their widespread use. We conducted a systematic review and meta-analysis to provide a quantitative assessment of their effect on measures of MM in individuals with, or without, T2DM. Methods. EMBASE, Medline, Pubmed, CINAHL, CENTRAL and Google Scholar were systematically searched for studies involving caloric restriction interventions up to 900 kilocalories per day reporting any measure of MM, in addition to fat mass (FM) or body weight (BW). Results. Forty-nine studies were eligible for inclusion, involving 4785 participants. Individuals with T2DM experienced significant reductions in MM (WMD −2.88 kg, 95% CI: −3.54, −2.22; *p* < 0.0001), although this was significantly less than the reduction in FM (WMD −7.62 kg, 95% CI: −10.87, −4.37; *p* < 0.0001). A similar pattern was observed across studies involving individuals without T2DM. MM constituted approximately 25.5% of overall weight loss in individuals with T2DM, and 27.5% in individuals without T2DM. Subgroup analysis paradoxically revealed greater BW and FM reductions with less restrictive interventions. Conclusions. Our review suggests that caloric restriction interventions up to 900 kilocalories per day are associated with a significant reduction in MM, albeit in the context of a significantly greater reduction in FM. Furthermore, MM constituted approximately a quarter of the total weight loss. Finally, our data support the use of less restrictive interventions, which appear to be more beneficial for BW and FM loss.

## 1. Introduction

Very-low-calorie diets (VLCDs) are interventions in which total energy intake is restricted to less than 800 kilocalories (kcal) per day, whilst low-calorie diets (LCDs) involve caloric restriction between 800 kcal and 1200 kcal per day. Both interventions lead to significant weight loss and improvement in outcomes in a variety of clinical contexts [1,2,3,4,5,6,7,8].

VLCD has received particular interest recently as a strategy for managing type 2 diabetes (T2DM). T2DM arises due to a toxic interaction between resistance to the actions of insulin and failure of adequate insulin secretion from pancreatic beta cells [9]. These two pathogenetic processes act synergistically, ultimately resulting in chronic hyperglycaemia [10], and the multitude of microvascular and macrovascular complications characteristic of manifest T2DM. The ability of a VLCD to improve both insulin sensitivity (IS) and beta-cell function (BCF) was recognised as far as four decades ago [11,12,13], making it an attractive option for the clinical management of T2DM. However, it was only more recently when the potential for a VLCD to induce remission of T2DM was realised [14].

Following further work by this and other groups [15,16,17] supporting the concept of VLCD-induced T2DM remission, the DiRECT study was conducted, establishing the potential of inducing diabetes remission through a large-scale intervention based on VLCD principles [18]. Notably, the intervention delivered in the DiRECT study was 825–853 kcal per day [18], and therefore would not strictly be described as a VLCD. Nonetheless, sustained diabetes remission was observed in 46% of the intervention group [18]. Furthermore, longer-term follow-up studies have demonstrated that this remission persists in a proportion of the participants [19,20].

Consequently, in 2018, the United Kingdom National Health Service (NHS) endorsed the use of the programme outlined in the DiRECT study for the management of T2DM in England [21]. Very recently, an analysis of the real-world application of the programme has provided further evidence of its efficacy to induce diabetes remission, albeit with a lower rate of success than demonstrated in the research context [22]. These findings have firmly validated the role of caloric restriction interventions in T2DM management, and their use is likely to become more commonplace in the future.

One considerable drawback of weight-loss programmes is the loss of skeletal muscle mass (MM) induced by these interventions [23]. This is particularly pertinent to individuals with T2DM as they experience accelerated age-related loss of skeletal muscle [24], leading to greater risk of frailty and disability than matched individuals without T2DM [25]. Furthermore, skeletal muscle is a major organ in the regulation of glucometabolic status, being responsible for over 85% of peripheral glucose disposal under hyperinsulinaemic conditions [26] and 70% of post-prandial glucose absorption [27]. Lower skeletal MM has been associated with worse glycaemic control [28]; therefore, rapid weight regained following the cessation of the intervention risks patients being left with a worse outcome overall.

This potential for deterioration in glycaemic control, as well as the increased risk of frailty and disability associated with the loss of MM, has led to significant concerns regarding the safety of implementing caloric restriction interventions on a widespread basis. Importantly, whilst reviews of this topic have suggested significant loss of MM following a VLCD [29,30], these have been narrative in nature with no quantitative synthesis or statistical analysis of the available evidence. Moreover, no review has specifically examined the impact of caloric restriction on skeletal muscle in individuals with T2DM. Given the increasing use of these interventions in this population, their accelerated loss of skeletal muscle and the potentially deleterious glycaemic and musculoskeletal effects of further exacerbating this process, this gap in the literature requires addressing.

We therefore undertook this systematic review and meta-analysis to examine the effects of interventions involving caloric restriction up to 900 kcal on measures of MM in individuals with T2DM. We also included an analysis of these effects in individuals without T2DM to provide some comparison and determine whether T2DM confers a greater risk of excessive MM loss.

## 2. Methods

This systematic review was conducted in accordance with the 2020 Preferred Reporting Items for Systematic Reviews and Meta-Analysis (PRISMA) guidance (see PRISMA checklist, Appendix A). The methodology and search strategy were outlined in a protocol prior to commencing the review and published in the International Prospective Register of Systematic Reviews (PROSPERO 2022), registration number CRD42024542940.

### 2.1. Search Strategy

A systematic search was performed for relevant studies published on the EMBASE, Medline, PubMed, CINAHL and Cochrane Central Register of Controlled Trials (CENTRAL) databases. The searches were performed on the 9 May 2024 and any article published from inception until that date meeting the eligibility criteria was included. A search of Google Scholar was also performed on the 10 May 2024 to examine for any relevant grey area literature, and any eligible articles identified within the first 10 pages of results were also included.

The following combination of terms was utilised for the overall search strategy: [(obes*) OR (diabet*) OR (overweight)] AND [(low calorie diet) OR (low energy diet) OR (hypocaloric diet) OR (meal replacement) OR (total diet replacement)] AND [(body composition) OR (lean body mass) OR (muscle mass) OR (fat free mass) OR (fat mass)]. These terms were adjusted for each specific database if required, and in the case of EMBASE and Medline, terms were mapped to EMTREE or MeSH subject headings where appropriate. Examples of the search strategy used in specific databases can be found in the Appendix A.

### 2.2. Study Eligibility Criteria

Studies were included if they were full articles of an interventional design published in a peer-reviewed journal, involving a VLCD or LCD up to 900 kcal per day, for at least 4 weeks. Both randomised controlled trials (RCTs) and non-randomised studies were eligible for inclusion. We decided to include LCD interventions up to 900 kcal as the UK NHS programme uses an 825–853 kcal intervention; thus, extending the calorie limit makes the findings more relevant to clinical practice. Furthermore, a meta-analysis comparing strict VLCDs to low-calorie liquid formula diets > 800 kcal/day observed no significant difference between the amount of weight loss in individuals with T2DM [31]. The meta-analyses were sub-grouped by caloric restriction, in order to identify and account for any potential heterogeneity that could have been introduced by the inclusion of these studies. For the remainder of this review, all included interventions will be described as VLCD.

Included studies were required to report a measure of muscle mass, e.g., fat-free mass (FFM), lean body mass (LBM) or skeletal muscle mass (SMM). Fat mass (FM) and body weight (BW) were listed as secondary outcomes, and data regarding these measures were extracted, although it was not a requirement of studies to report them. At least one experimental group in which VLCD was the only intervention was necessary, and study groups were only included if the mean age was ≥40 years and mean BMI ≥ 25 kg·m^−2^. Studies were excluded if they investigated individuals with type 1 diabetes, or pregnant, neonatal, paediatric and adolescent populations. Studies regarding very-low-calorie ketogenic diet interventions, or if the VLCD occurred alongside a concurrent formal exercise programme, were also excluded. Finally, case reports, conference proceedings or articles published in a non-English language were not eligible for inclusion.

### 2.3. Study Selection

Following conduction of searches and removal of duplicates, the titles and abstracts of the results were independently screened by four authors (A.B., S.K., O.O. and R.R.). Any non-interventional article that broadly met the inclusion criteria (such as a review) was identified as a secondary source. The reference lists of secondary sources were then evaluated by another author (O.A.) in order to identify further potentially eligible studies. The full texts of all citations from the initial and secondary searches were obtained and independently assessed for eligibility by the same authors, and all articles deemed to meet the stated criteria were included.

### 2.4. Data Extraction

Required data from all included studies were extracted and collated into a single spreadsheet (Microsoft Excel Version 2409). In studies containing more than one intervention group that could be included in the review, data were obtained for each cohort. Information regarding the study type, intervention duration, daily caloric restriction and method of body composition assessment were collected for included studies. Additionally, participant number and characteristics (gender, mean age, mean BMI and diabetes status) were extracted for each included intervention cohort.

For the primary and secondary outcomes, both pre-intervention and post-intervention measures were recorded. If either of these values were not immediately available from the article or any published Appendix A, the corresponding authors were contacted to request the missing data. If despite this the data were not obtained, then the results of these studies were discussed in the narrative synthesis only. Alternatively, if the data were presented graphically within the article, Plot Digitizer online software (https://plotdigitizer.com/, accessed on 5 July 2024) was used to provide an estimate of the missing values, in accordance with Cochrane guidance [32].

### 2.5. Quality Assessment

Risk of bias was assessed using the Cochrane Risk of Bias 2 (ROB2) tool for RCTs [33,34]. Each study was attributed a rating of “low”, “some concerns” or “high risk of bias” based on appraisal of five bias domains [33].

In the case of single-group or multiple-group non-randomised interventional studies, the Newcastle–Ottawa Scale (NOS) was utilised, which provides an evaluation of overall study quality based on the responses to eight questions divided into three domains [35]. Stars were allocated depending on the responses to these questions and a study was deemed “high quality” if it received 7–9 stars, “fair quality” if 4–6 stars and any study receiving fewer than 4 stars was rated as “low quality” overall. For single-group studies, as two questions were not relevant to these types of studies, the criteria for each rating were altered as follows: “high quality” 6–8 stars, “fair quality” 3–5 stars, “low quality” < 3 stars.

In an analysis where 10 or more studies were included, the presence of publication bias was assessed via generation of a funnel plot for visual inspection of the distribution of results and performance of Egger’s regression test.

### 2.6. Statistical Analysis

Statistical analysis was performed using the StataSE version 18.5 (StataCorp, College Station, TX, USA) software package. The weighted mean difference (WMD) was calculated using the pre-intervention and post-intervention mean and standard deviation (SD) from each included cohort. This was appropriate as all measures were reported in kilograms. Calculations were performed using the random-effects Hedges model, as we anticipated a potential for significant heterogeneity due to variations in caloric restriction and population characteristics. Statistical significance was defined as *p* < 0.05 and data are presented as WMD with 95% confidence interval (CI).

In cases where standard error of the mean (SEM) or 95% CI was the reported variance value in the article, SD was calculated from these as advised by Cochrane guidance [36]. In the situation where only a pre- or post-intervention measure was reported alongside the mean change in the outcome, the required information was requested. If this was not received, then the missing measure was calculated using available data, and the corresponding SD was assumed to be equal to that of the reported measure. Sensitivity analysis was performed to determine whether inclusion of these studies and their calculated values significantly altered the observed effect.

Studies were divided depending on the glycaemic status of the intervention groups with separate analysis performed for studies including individuals with T2DM (including prediabetes) and those involving individuals without T2DM. This enabled comparison of the effect of a VLCD on muscle mass between these two populations to establish if differential effects exist. In cases where the cohort was mixed, the predominant status of the participants was established, and the study was included in the corresponding analysis.

Forest plots were generated for each set of populations, with the analysis of individuals without T2DM (NDM analysis) further divided into subgroups based on the level of caloric restriction. Subgroups were tested for significant differences if ≥2 data points were present. Each forest plot of change in MM measures was presented alongside the corresponding forest plot for FM or BW changes, in order to enable direct comparisons between the outcome measures, providing greater context to the results.

In addition to the sensitivity analysis described above, additional sensitivity analyses were performed to determine if the reported measure of MM, or the method of assessing body composition, significantly affected the observed findings. Study heterogeneity was assessed with visual examination of the generated forest plots, alongside *I*^2^ and Tau statistics (interpreted in accordance with Cochrane guidance).

## 3. Results

### 3.1. Study Characteristics

An overview of the study selection process is summarised in Figure 1. Primary search of the databases and Google Scholar yielded a total of 4378 citations, which was reduced to 2056 following the removal of duplications. Screening of titles and abstracts of these citations resulted in the exclusion of 1360; thus, 696 full texts were reviewed to determine eligibility for inclusion. In addition, 50 potentially eligible citations were identified from the secondary source review, and the full texts of these articles were also assessed. Of the 746 articles that were reviewed, 49 were deemed to have met the inclusion criteria and were therefore included in the review. Of this number, 43 were included in the quantitative synthesis, as 5 did not report data in a format that was amenable to this and 1 did not explicitly state the daily caloric limit imposed upon participants. The details of the 49 studies included in the review are outlined in Table 1.

Thirty-two articles reported the outcomes of single-group interventional studies, forming the vast majority of the included studies. The remaining studies comprised 10 RCTs and 7 multiple-group non-randomised interventional studies. Several studies were RCTs by design; however, all participants followed an initial VLCD period prior to randomisation to a subsequent intervention. As all participants acted as one intervention group during the period of interest, these were classified as single-group interventional studies for the purposes of this review. Eight studies contained more than one cohort that was eligible for inclusion in the review, with each cohort being analysed as a separate entity. Thus, 51 eligible cohorts were included in the meta-analysis.

A total of 4555 participants were involved in the studies included in this review. The caloric restriction ranged from 430 kcal to 900 kcal per day, and the intervention length ranged from 4 to 16 weeks. In one study, the intervention was a 33% caloric restriction, which was described as a VLCD. As the specific daily calorie intake limit could not be quantified, this study was not included in the meta-analysis and was therefore reported in the narrative synthesis only. Nine studies involved participants with T2DM or prediabetes, whilst the remaining forty included individuals without T2DM. Nine study cohorts involving 2422 participants were included in the T2DM meta-analysis, while forty-five cohorts involving 2006 participants were included in the NDM meta-analysis.

### 3.2. T2DM Analysis

The results of the T2DM meta-analysis are presented graphically in Figure 2 and Figure 3. All but one study [14] involved interventions with daily calorie limits between 800 and 900 kcal and therefore subgroup analysis by caloric restriction was not performed.

Among the included studies, the interventions induced a significant 2.94 kg reduction in MM (95% CI: −3.59, −2.29; *p* < 0.0001). However, this was significantly less than the 8.10 kg reduction in FM (95% CI: −9.61, −6.58; *p* < 0.0001). A similar reduction in MM was observed in the studies reporting both MM and BW changes. MM reduced by 2.89 kg (95% CI: −3.52, −2.26; *p* < 0.0001) in the context of a 11.32 kg (95% CI: −12.43, −10.22; *p* < 0.0001) weight loss. Thus, 25.5% of the BW reduction experienced by the participants in these studies comprised MM.

*I*^2^ values in the BW and both MM forest plots were 0.00%, suggesting minimal heterogeneity within these data sets. However, the *I*^2^ value of 24.5% was suggestive of significant heterogeneity within the FM analysis. From visual inspection of the forest plots, it can be appreciated that while all studies reported reductions in FM, the reductions in the study by Ivan et al. [77] were uncharacteristically low for both FM and BW. Review of the methodology employed by this study revealed that participants were allowed to independently prepare their intervention diets, rather than total diet replacement more typically used in a VLCD. Thus, the minimal BW and FM reduction may reflect issues with compliance to this study’s intervention, which may explain the observed heterogeneity.

One study was not amenable for inclusion in the quantitative synthesis but reported a significant 2.9 kg reduction in MM, in the context of 8.2 kg weight loss over 4 weeks of an 800 kcal per day VLCD [74]. Of interest, the study by Athithan et al. [81] reported changes in the LBM and FM percentage, with participants grouped according to ethnicity. The LBM percentage significantly increased following the 12-week intervention in both cohorts, alongside significant reductions in the FM percentage. Furthermore, there were no significant differences between the changes in each cohort, suggesting that changes in body composition are consistent across ethnic groups [81].

In summary, the studies included in this analysis suggest that significant loss of MM occurs with a VLCD in individuals with T2DM. However, these losses are significantly less than the reduction in FM and comprise approximately 25% of the overall weight loss, although there is some notable variation in this. As the extent of MM reduction is significantly lower than the corresponding amount of FM, the MM percentage experiences a corresponding increase, and this finding appears to be consistent across different ethnicities.

### 3.3. NDM Analysis

The forest plots generated from the NDM analyses are presented in Figure 4 and Figure 5. As observed in the T2DM analysis, MM reduced significantly by 2.77 kg (95% CI: −3.29, −2.24; *p* < 0.0001) and this reduction was significantly lower than that observed in FM (WMD −6.91 kg, 95% CI: −7.41, −6.41; *p* < 0.0001). This analysis was sub-grouped by level of caloric restriction with 20 cohorts included in the 400–600 kcal per day group, 16 cohorts included in the 600–800 kcal per day group and the remaining 8 cohorts in the 800–900 kcal per day group. No significant differences were observed between the MM changes in any subgroup (test for group differences *p* = 0.50). However, the test for group differences was significant in the FM analysis (*p* = 0.01), with the 400–600 kcal per day subgroup demonstrating lower reductions in FM (WMD −6.32 kg, 95% CI: −7.62, −5.01) than the other subgroups (600–800 kcal WMD −8.78 kg, 95% CI: −9.83, −7.74; 800–900 kcal WMD −7.18 kg, 95% CI: −8.26, −6.11).

In the comparison with BW, a VLCD induced a −2.73 kg (95% CI: −3.26, −2.29; *p* < 0.0001) change in MM, alongside a −9.92 kg (95% CI: −10.67, −9.16; *p* < 0.0001) change in BW (Figure 5). Thus, 27.5% of the total weight loss was attributable to MM, which was similar to the figure observed in the T2DM analysis. Overall, similar reductions in MM, BW and FM were observed across both T2DM and NDM analyses. Again, there was no significant difference in MM changes between the different levels of caloric restriction (test for group differences *p* = 0.69). However, in contrast, BW changes did differ significantly between the subgroups (test for group differences *p* < 0.05). The lowest BW reductions were again noted in the 400–600 kcal per day subgroup (WMD −8.62 kg, 95% CI: −10.24, −6.99), compared to −11.60 kg (95% CI: −13.25, −9.96) with the 600–800 kcal per day and −11.52 kg (95% CI: −13.25, −9.78) with 800–900 kcal per day interventions.

Visual inspection of both MM forest plots and analysis of *I*^2^ values again suggested minimal heterogeneity in MM changes across the included studies. However, the presence of significant heterogeneity was detected among the FM and BW forest plots, albeit confined to the 400–600 kcal subgroup in both cases. More detailed inspection revealed that the study by Rolland et al. [57] was likely to be the primary driver of this heterogeneity, due to the greater degree of BW and FM loss reported by this study. Notably, this study subjected participants to three months of a 550 kcal per day VLCD, which was longer than any other study in this subgroup. Thus, this methodological difference is the likely reason for the observed heterogeneity. There was also some variation in response noted upon visual inspection of the forest plots, which is likely to arise from additional clinical differences between the intervention populations of each study cohort.

Four studies were not included in the meta-analysis. In keeping with the findings of the meta-analysis, Soenen et al. [61] reported significant reductions in both FFM and FM following 6 weeks of 33% caloric restriction, associated with significant weight loss. Similar findings were observed by Nymo et al. [69] in response to 8 weeks of a VLCD in 31 individuals. Carella et al. [41] also reported numerical reductions in FFM following 12 weeks of a VLCD; however, there was no comment on the significance of these reductions, and there was no associated report of FM or BW change. Finally, Iepsen et al. [62] reported changes in the LBM and FM percentage in response to 8 weeks of an 810 kcal per day intervention. Similar to the findings of Athithan et al. [81] in individuals with T2DM, Iepsen et al. observed a significant reduction in the FM percentage, with a consequent significant rise in the LBM percentage [62]. Thus, the findings of these four studies appear to be consistent with the results of our analysis.

### 3.4. Sensitivity Analysis

In order to determine whether inclusion of studies with incomplete data had a significant effect on the review results, sensitivity analysis was performed. Repeat analysis of the data was performed excluding studies that required estimation of post-intervention mean and SD, or those that required estimation of the outcomes from graphs within the published article. Seven studies included in the meta-analysis did not report the full data required [39,45,60,63,64,65,79]. As reported in Appendix A, removal of these studies did not significantly alter the observed effects.

We also performed sensitivity analysis to determine whether the method of body composition assessment used in each study, or the measure of MM reported by studies, could have had a significant interaction with the overall result. As outlined in Appendix A, neither of these factors demonstrated any differences in the results with any method or measure (test for group differences, *p* = 0.33 and *p* = 0.32, respectively).

### 3.5. Study Quality Assessment

The results of the quality assessments are reported in the Appendix A. All but two of the single-group interventional studies received six stars using the NOS. These two studies [48,79] experienced a high number of participants dropping out of the study with no clear attempt to account for the dropouts and any potential bias this may have introduced into their results. Nevertheless, all studies received a general “high quality” rating overall. Similarly, all multiple-group cohort studies received 8 stars, corresponding to a “high quality” overall rating.

Four out of the ten included RCTs received an overall low risk of bias rating using the Cochrane ROB2 tool. Five RCTs [45,61,65,67,70] were deemed to have some concerns, primarily due to lack of information regarding whether the allocation sequence was concealed until participant recruitment and assignment to their respective intervention. Additionally, in two studies [67,70], there were concerns noted regarding potential deviations from the intended intervention protocol. One study [57] was rated high risk of bias, due to the presence of significant differences in the baseline characteristics between the intervention groups, raising the suspicion of an issue with the randomisation process.

### 3.6. Publication Bias

Fewer than 10 studies were included in the T2DM analysis; therefore, publication bias was only assessed within the studies included in the NDM analysis. No evidence of publication bias was detected using either assessment. The funnel plot (Appendix A) demonstrated good symmetry around the mean value and the calculated Egger regression test was non-significant (*p* = 0.78).

## 4. Discussion

VLCD-based interventions are increasing in popularity and an 825–853 kcal per day meal replacement programme is currently recommended for the treatment of T2DM in the UK [85,86]. However, concerns regarding potential significant loss of MM associated with a VLCD and similar interventions [87,88,89,90] in clinical populations persist. Despite other comprehensive reviews on this topic [29,30,91], no quantitative synthesis of the available evidence specific to VLCD-induced changes has been conducted. Furthermore, the effect of VLCD-based interventions in individuals with T2DM specifically has never been determined, despite this population being most likely to receive these interventions as part of their routine care.

This systematic review and meta-analysis showed that interventions restricting daily energy intake to 900 kcal or less cause significant reductions in MM, in both individuals with T2DM and those without T2DM. Expectedly, significant reductions in BW occurred, and our results demonstrated that approximately 25.5% and 27.5% of the total weight loss was attributable to MM among individuals with and without T2DM, respectively. In keeping with this, there was a significantly greater absolute reduction in FM than MM in our included studies. This would result in a significant increase in relative MM percentage following the intervention, which was corroborated by some of the studies discussed in the narrative synthesis [62,81]. Perhaps most notably, the demonstrated reductions in MM were similar, regardless of the glycaemic status of the included participants.

This lattermost point is particularly relevant given the aforementioned liquid energy diet programme ongoing in the UK. Following positive results in the recently published real-world evaluation of the programme [22], it is likely that this will continue for the medium to long term. However, the increased rate of age-related muscle loss associated with T2DM [92] understandably causes significant apprehension regarding the use of these types of interventions in this population [93]. The findings of this review suggest that although VLCD-based interventions do cause a significant reduction in MM, the concurrent presence of T2DM does not appear to further exacerbate this. This is a novel finding and may provide reassurance to clinicians involved in the management of individuals with T2DM.

The proportion of weight loss attributable to changes in MM was remarkably similar to the 25% figure reported by Willoughby et al. [94], suggesting that our findings are robust. Chaston et al. also observed similar values when examining the FFM proportion of weight loss across multiple weight-loss interventions [29]. This compares favourably to the 30–33% proportion of total weight loss observed within the first 3 months following bariatric surgery reported in a recent meta-analysis [95]. It is, however, important to note that the concurrent weight loss observed over this time period was not reported, and associations between the proportion of MM loss and rate of BW reduction have been suggested [29].

Although caloric restriction regimes between 800 and 900 kcal per day are not strictly considered as a VLCD, the UK NHS programme employs an 825–853 kcal per day caloric restriction; thus, inclusion of these studies made our findings more clinically relevant. Intriguingly, we did not observe any significant difference in the weight loss, and associated change in body composition, between the 800–900 kcal per day and 600–800 kcal per day subgroups. This is in line with the findings of Leslie et al. [31], providing further strength to the argument against the use of unnecessarily restrictive VLCD interventions.

Furthermore, our findings appeared to suggest a significant weight-loss benefit associated with the higher calorie limit studies, compared to the 400–600 kcal per day interventions. Notably, similar assertions have been made in guidance published over 30 years ago regarding the use of a VLCD for the treatment of obesity [87]. Whilst this appears counterintuitive, this is likely to reflect improved compliance, as well as greater ability to sustain the intervention for longer [87]. Indeed, the interventions in the most restrictive subgroup were largely between 4 and 6 weeks’ duration, whilst those of the other two subgroups tended to be conducted over 7 weeks or more. Despite the increased weight loss, MM change did not significantly differ, providing further benefit to the use of less restrictive caloric restriction regimes.

Although our study has demonstrated significant reductions in MM associated with a VLCD, we were unable to relate these to changes in muscle strength or other clinically important outcomes such as disability or frailty. This is an important consideration as recent evidence suggests that muscle strength may be more closely related to the development of T2DM than muscle quantity [96,97]. Unfortunately, evidence regarding the effect of VLCD-induced MM loss on muscle strength is lacking, particularly in the context of T2DM. However, one meta-analysis has suggested that a VLCD causes significant reductions in knee extensor strength, although there was no concurrent reduction in hand-grip strength [98]. The authors also highlighted a potential relationship between changes in MM and these measures of muscle strength [98]. Thus, in light of our findings, the associated changes in muscle strength following VLCD use in T2DM require further investigation, particularly since individuals with T2DM are already at risk of lower muscle strength and greater disability [25,99].

In relation to this, cessation of VLCD interventions is typically followed by significant weight regain [100,101]. It has been suggested that the majority of the BW regained is FM, which could subsequently result in a net reduction in MM, despite BW returning to baseline [102]. This could eventually stimulate deleterious effects on overall metabolic health [102]. In contrast with these assertions, the several studies included in this review that reported post-VLCD changes in body composition largely demonstrated that increases in both FM and MM occur [51,65,67,71,83]. Notably, none of these studies were conducted in individuals with T2DM who are known to exhibit increased levels of ectopic fat deposition [103]; thus, they may be prone to greater accumulation of fat during post-VLCD weight regain. This represents another area that requires further review.

One major limitation of this review was the lack of high-quality RCTs investigating the effect of a VLCD on body composition. It is important to recognise that the nature of VLCD interventions generates difficulties in the blinding of participants and investigators to the allocated intervention. However, RCTs can still be conducted to investigate the efficacy of this intervention despite these challenges. The majority of articles included in this review were of single-cohort interventional study design. These are inherently prone to bias, although body composition is typically measured via automated processes, which may limit the level of bias that can be introduced into the results of such studies. The initial scoping review conducted during development of the protocol revealed a scarcity of RCTs; thus, we decided to include all types of studies, in order to provide some insight into this topic. Conduction of more high-quality RCTs will enable the performance of a much more robust systematic review to address this question.

Additionally, a few studies reported incomplete data, precluding their inclusion in the meta-analysis. Where possible, missing data were calculated from the available data, which contributed to the overall findings. Of note, our sensitivity analysis revealed that exclusion of such studies from the meta-analysis did not significantly alter our results. Significant heterogeneity was detected in some of the analyses performed, primarily resulting from methodological differences between the included studies, along with some likely clinical differences within the studied intervention populations. Finally, as previously mentioned, this review did not examine changes in strength or risk of disability, which is crucial to understanding the effect of these observed changes on the lives of individuals undergoing these interventions.

In conclusion, we have demonstrated that in individuals with T2DM, VLCDs stimulate significant reductions in MM, which comprise approximately 25.5% of the total weight loss, similar to that observed in individuals without T2DM. The presence of T2DM therefore does not exacerbate the loss of MM during these interventions. We observed greater weight loss with the less restrictive interventions, without significant changes in MM loss, suggesting that 800–900 kcal per day is adequate for maximising weight loss. Further studies are required to assess the potential impact of this MM loss on strength, physical function and long-term weight maintenance/regain, particularly in individuals with T2DM in which evidence is severely lacking. We hope the findings of this review will be of value to clinicians involved in the care of individuals with T2DM when making decisions regarding the use of caloric restriction interventions for their management.

## Figures and Tables

**Figure 1 nutrients-16-03328-f001:**
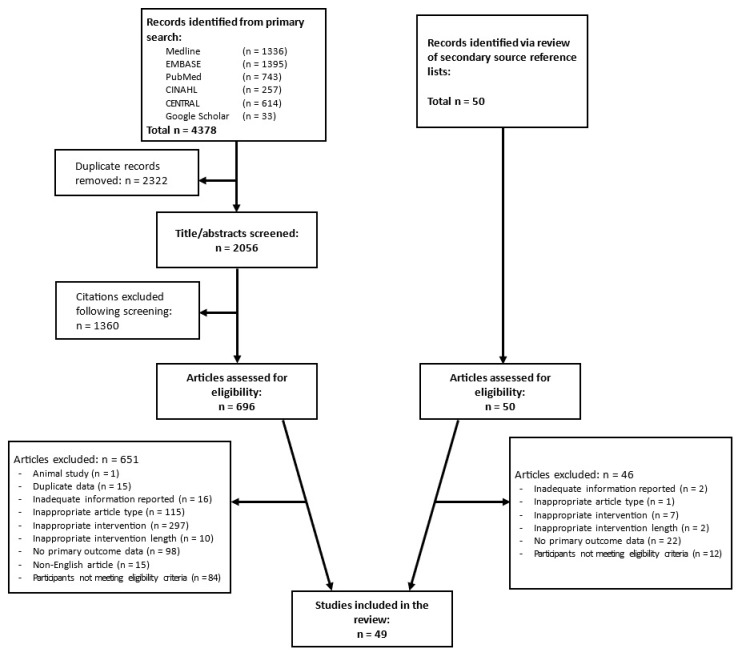
Study selection process overview.

**Figure 2 nutrients-16-03328-f002:**
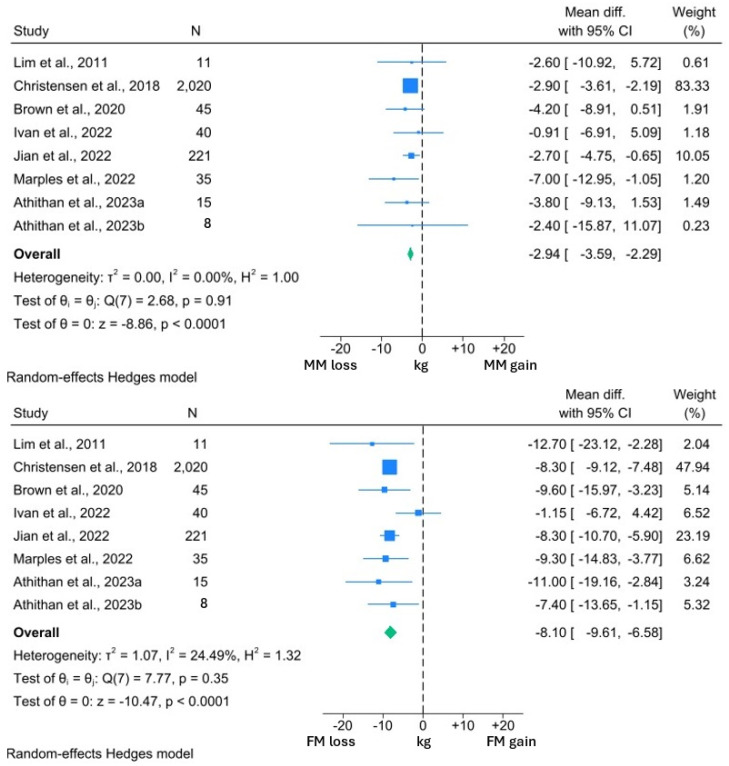
Forest plots showing comparison between change in MM and FM in individuals with T2DM [14,68,72,77,78,79,81].

**Figure 3 nutrients-16-03328-f003:**
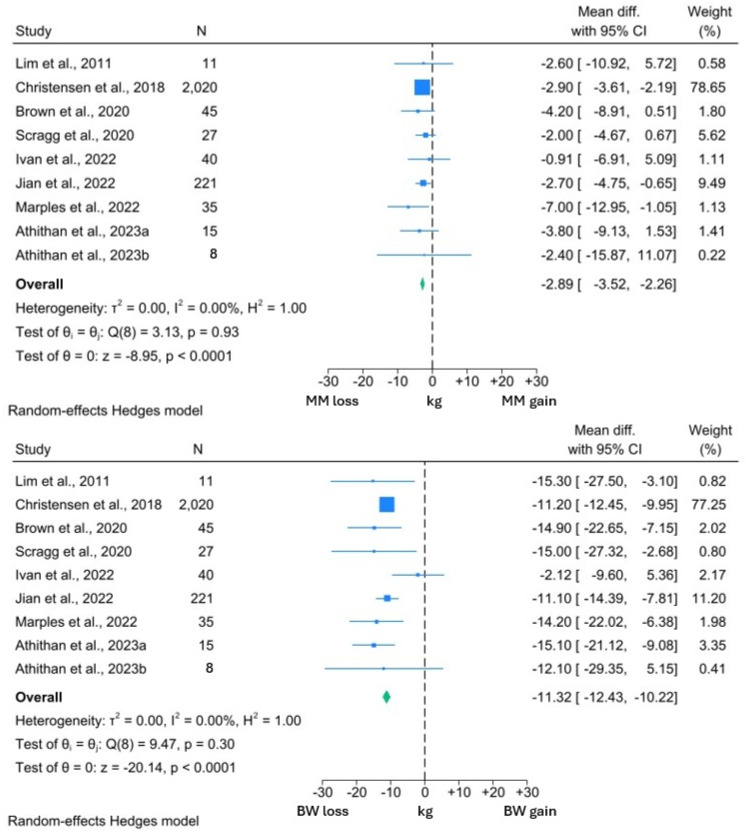
Forest plots showing comparison between change in MM and BW in individuals with T2DM [14,68,72,73,77,78,79,81].

**Figure 4 nutrients-16-03328-f004:**
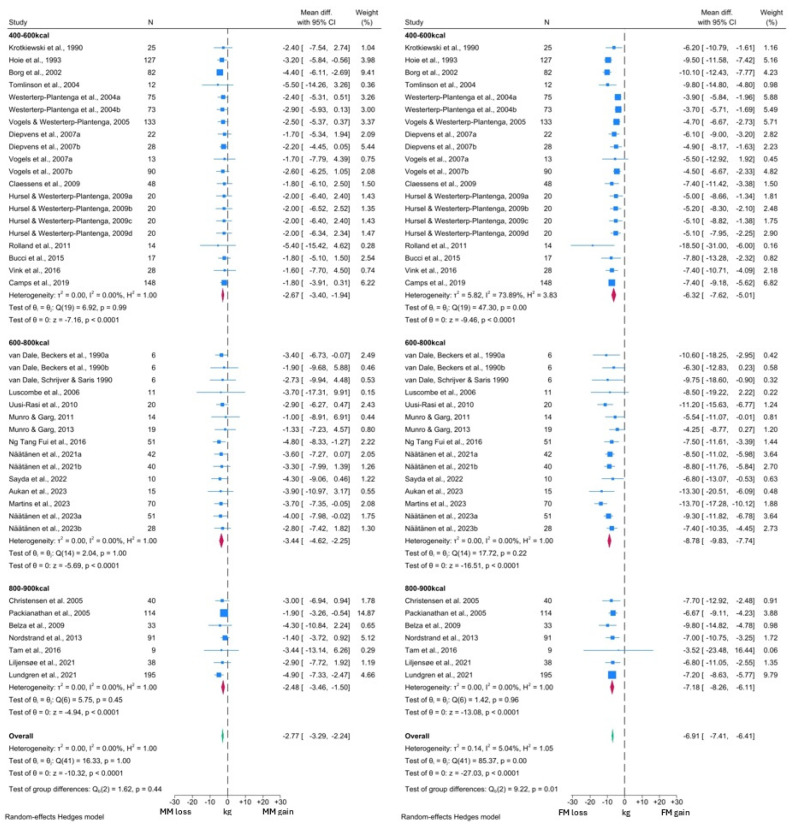
Forest plots showing comparison between change in MM and FM in individuals without T2DM [37,38,39,40,42,43,44,45,46,47,49,50,51,52,53,54,55,56,57,59,60,63,64,65,66,70,71,75,76,80,82,83,84].

**Figure 5 nutrients-16-03328-f005:**
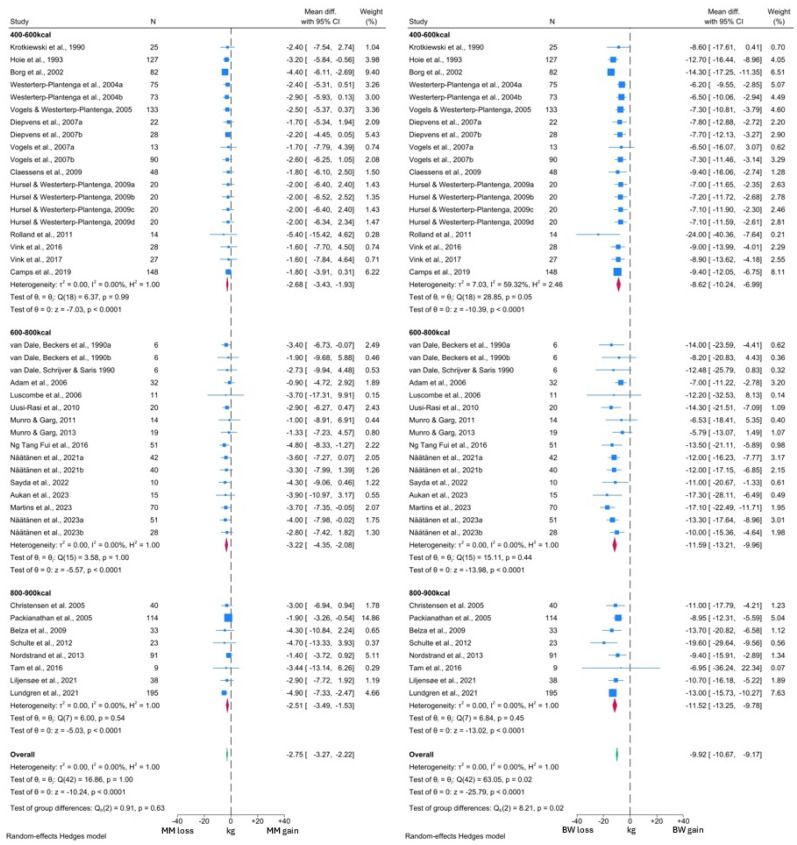
Forest plots showing comparison between change in MM and BW in individuals without T2DM [37,38,39,40,42,44,45,46,47,48,49,50,51,52,53,54,55,56,57,58,59,60,63,64,65,67,70,71,75,76,80,82,83,84].

**Table 1 nutrients-16-03328-t001:** Details of included studies. RCT—randomised controlled trial, SGCS—single-group interventional studies, MGCS—multiple-group interventional studies (non-randomised), IQR—interquartile range, BIA—bioimpedance analysis, DXA—dual-energy X-ray absorptiometry, ADP—air displacement plethysmography. * denotes SEM, ^$^ average age for whole study population.

Study	Study Design	Daily Caloric Restriction	Intervention Duration	Sample Size (Number of Females)	Diabetes Status	Age Mean (SD)	Baseline BMI Mean (SD)	Method of Body Composition Assessment	Outcomes of Interest Reported
Krotkiewski et al., 1990 [37]	SGCS	544 kcal	4 weeks	25 (all female)	NDM	40.1 (2.92) *	36.9 (1.2) *	Potassium estimation	LBM, FM, weight
van Dale, Beckers et al., 1990a [38]	MGCS	3.0 MJ	4 weeks	6 (all female)	NDM	41 (2.4) *	33.2 (1.0) *	Body density calculation	FFM, FM, weight
van Dale, Beckers et al., 1990b [38]	MGCS	3.0 MJ	4 weeks	6 (all female)	NDM	41 (1.9) *	33.5 (0.8) *	Body density calculation	FFM, FM, weight
van Dale, Schrijver & Saris 1990 [39]	MGCS	3.0 MJ	5 weeks	6 (all male)	NDM	41.0 (4.0) *	32.0 (2.0) *	Hydrostatic weighing	FFM, FM, weight
Hoie et al., 1993 [40]	SGCS	430 kcal	8 weeks	127 (82)	NDM	41.2 (range 18–72)	33.2 (range 25–51)	Infra-red technology	LBM, FM, weight
Carella et al., 1997 [41]	SGCS	800 kcal	12 weeks	37 (24)	NDM	42 (8.2)	Female—37 (6) Male—43 (4)	BIA	FFM, FM, weight
Borg et al., 2002 [42]	SGCS	500 kcal	2 months	82 (all male)	NDM	42.6 (4.6)	32.9 (2.6)	Hydrostatic weighing	FFM, FM, weight
Tomlinson et al., 2004 [43]	SGCS	559 kcal (male)/425 kcal (female)	10 weeks	12 (6)	NDM	49 (range 23–58)	35.9 (0.9) *	DXA	LBM, FM, weight
Westerterp-Plantenga et al., 2004a [44]	SGCS	2.1 MJ	4 weeks	75 (gender undisclosed)	NDM	43.8 (10.1)	29.3 (2.5)	Deuterium dilution	FFM, FM, weight
Westerterp-Plantenga et al., 2004b [44]	SGCS	2.1 MJ	4 weeks	73 (gender undisclosed)	NDM	44.5 (10.5)	29.7 (2.6)	Deuterium dilution	FFM, FM, weight
Christensen et al., 2005 [45]	RCT	3.4 MJ	8 weeks	40 (35)	NDM	60.5 (11.6)	36.3 (5.6)	BIA	LBM, FM, FFM
Packianathan et al., 2005 [46]	SGCS	900 kcal	16 weeks	114 (all female)	NDM	48.5 (8.25)	36.1 (5.62)	ADP	LBM, FM, weight
Vogels & Westerterp-Plantenga, 2005 [47]	SGCS	2.1 MJ	6 weeks	133 (gender undisclosed)	NDM	48.1 (9.5)	31.1 (3.7)	Deuterium dilution	FFM, FM, weight
Adam et al., 2006 [48]	SGCS	2540 kJ	6 weeks	32 (23)	NDM	44 (9)	30.1 (2.6)	Deuterium dilution	FFM, weight
Luscombe et al., 2006 [49]	SGCS	3300 kJ	8 weeks	11 (5)	NDM	50 (3) *	34.0 (1.7) *	DXA	LBM, FM, weight
Diepvens et al., 2007a [50]	SGCS	2.1 MJ	6 weeks	22 (all female)	NDM	40.3 (9.7)	28.9 (1.7)	Deuterium dilution	FFM, FM, weight
Diepvens et al., 2007b [50]	SGCS	2.1 MJ	6 weeks	28 (all female)	NDM	41.2 (9.3)	28.5 (2.2)	Deuterium dilution	FFM, FM, weight
Vogels et al., 2007a [51]	SGCS	500 kcal	6 weeks	13 (gender undisclosed)	NDM	48.1 (9.5) ^$^	33.7 (4.7)	Deuterium dilution	FFM, FM, weight
Vogels et al., 2007b [51]	SGCS	500 kcal	6 weeks	90 (gender undisclosed)	NDM	48.1 (9.5) ^$^	30.5 (3.5)	Deuterium dilution	FFM, FM, weight
Belza et al., 2009 [52]	SGCS	3.4 MJ	8 weeks	33 (17)	NDM	43.0 (10.5)	34.0 (3.1)	BIA	FFM, FM, weight
Claessens et al., 2009 [53]	SGCS	500 kcal	5–6 weeks	48 (31)	NDM	46.0 (2.2)/45.4 (2.2)/44.9 (2.0) *	32.4 (1.2)/32.9 (1.6)/33.4 (1.0) *	Hydrostatic weighing	FFM, FM, weight
Hursel & Westerterp-Plantenga, 2009a [54]	SGCS	2.1 MJ	4 weeks	20 (11)	NDM	44 (2) ^$^	29.6 (2.1)	Deuterium dilution	FFM, FM, weight
Hursel & Westerterp-Plantenga, 2009b [54]	SGCS	2.1 MJ	4 weeks	20 (11)	NDM	44 (2) ^$^	29.5 (2.0)	Deuterium dilution	FFM, FM, weight
Hursel & Westerterp-Plantenga, 2009c [54]	SGCS	2.1 MJ	4 weeks	20 (11)	NDM	44 (2) ^$^	29.6 (2.1)	Deuterium dilution	FFM, FM, weight
Hursel & Westerterp-Plantenga, 2009d [54]	SGCS	2.1 MJ	4 weeks	20 (11)	NDM	44 (2) ^$^	29.5 (1.9)	Deuterium dilution	FFM, FM, weight
Uusi-Rasi et al., 2010 [55]	SGCS	778 kcal	3 months	20 (gender undisclosed)	NDM	42.1 (3.7)	33.3 (3.3)	DXA	FFM, FM, weight
Lim et al., 2011 [14]	SGCS	600 kcal	8 weeks	11 (2)	T2DM	49.7 (2.5) *	33.6 (1.2) *	ADP	FFM, FM, weight
Munro & Garg, 2011 [56]	SGCS	3000 kJ	4 weeks	14 (3)	NDM	42 (2.0)	33.04 (3.17)	BIA	FFM, FM, weight
Rolland et al., 2011 [57]	RCT	550 kcal	3 months	14 (9)	NDM	41.9 (6.5)	46.7 (9.0)	BIA	FFM, FM, weight
Schulte et al., 2012 [58]	SGCS	800 kcal	12 weeks	23 (15)	NDM	42.8 (2.6) *	44.1 (1.1) *	BIA	LBM, FM, weight
Munro & Garg, 2013 [59]	RCT	3000 kJ	4 weeks	19 (15)	NDM	47.11 (2.05) *	33.70 (0.83) *	BIA	FFM, FM, weight
Nordstrand et al., 2013 [60]	MGCS	900 kcal	7 weeks	91 (57)	NDM	42.3 (9.6)	45.7 (5.5)	BIA	SMM, FM, weight
Soenen et al., 2013 [61]	RCT	33% CR	6 weeks	36 (24)	NDM	44 (4)	32 (0.5)	ADP	FFM, FM, weight
Iepsen et al., 2016 [62]	SGCS	810 kcal	8 weeks	20 (gender undisclosed)	NDM	43 (9.6)	33.5 (2.2) *	DXA	LBM, FM, weight
Ng Tang Fui et al., 2016 [63]	RCT	640 kcal	10 weeks	51 (gender undisclosed)	NDM	52.8 (IQR 47.6–60.1)	37.3 (IQR 34.7–41.6)	DXA	LBM, FM, weight
Tam et al., 2016 [64]	MGCS	800 kcal	8 weeks	9 (8)	Mixed (majority NDM)	45 (5) *	48.8 (3.2) *	DXA	FFM, FM, weight
Vink et al., 2016 [65]	RCT	500 kcal	5 weeks	28 (15)	NDM	50.7 (1.5) *	31.0 (0.4) *	ADP	FFM, FM, weight
Bucci et al., 2015 [66]	SGCS	2.3 MJ	6 weeks	17 (12)	NDM	42 (6)	34.0 (3.9)	BIA	Muscle mass, FM, weight
Vink et al., 2017 [67]	RCT	500 kcal	5 weeks	28 (15)	NDM	50.8 (1.5) *	30.8 (0.4) *	ADP	FFM, FM, weight
Christensen et al., 2018 [68]	SGCS	3.4 MJ	8 weeks	2020 (1504)	Prediabetes	51.6 (11.6)	35.4 (6.6)	DXA	FFM, FM, weight
Nymo et al., 2018 [69]	SGCS	2.8 MJ (male)/ 2.3 MJ (female)	8 weeks	31 (13)	NDM	43 (10) *	36.7 (4.5) *	ADP	FFM, FM, weight
Liljensøe et al., [70]	RCT	810 kcal	8 weeks	38 (27)	Mixed (majority NDM)	65 (range 46–81)	31.6 (range 30.6–32.6)	DXA	LBM, FM, weight
Lundgren et al., 2021 [71]	SGCS	800 kcal	8 weeks	195 (124)	NDM	42 (12)	37.0 (2.9)	DXA	LBM, FM, weight
Brown et al., 2020 [72]	RCT	800–820 kcal	12 weeks	45 (25)	T2DM	58.5 (IQR 50.1–64.2)	36.6 (5.1)	BIA	LBM, FM, weight
Scragg et al., 2020 [73]	SGCS	800 kcal/day	8 weeks	27 (12)	Mixed (majority T2DM)	56 (12)	42 (8)	BIA	SMM, FM, weight
Behary et al., 2019 [74]	RCT	800 kcal	4 weeks	22 (12)	T2DM	47.0 (10.2)	39.1 (4.3)	BIA	Muscle mass, FM, weight
Camps et al., 2019 [75]	SGCS	2.1 MJ	8 weeks	148 (109)	NDM	41 (9)	31.9 (3.0)	ADP	FFM, FM, weight
Näätänen et al., 2021a [76]	SGCS	600 kcal	7 weeks	42 (30)	NDM	49.6 (9.5)	34.0 (2.3)	BIA	FFM, FM, weight
Näätänen et al., 2021b [76]	SGCS	600 kcal	7 weeks	40 (31)	NDM	49.1 (9.1)	34.3 (2.7)	BIA	FFM, FM, weight
Ivan et al., 2022 [77]	MGCS	800 kcal	30 days	40 (20)	T2DM	51.83 (1.8)	32.64 (0.98)	BIA	FFM, FM, weight
Jian et al., 2022 [78]	SGCS	810 kcal	8 weeks	221 (156)	T2DM	54 (95% CI 53–55)	34.1 (95% CI 33.3–34.9)	BIA	FFM, FM, weight
Marples et al., 2022 [79]	SGCS	825–853 kcal	12 weeks	35 (15)	T2DM	50.4 (10.5)	34.4 (4.5)	BIA	FFM, FM, weight
Sayda et al., 2022 [80]	SGCS	600–800 kcal	6 weeks	10 (all male)	NDM	45.9 (8.3)	32.2 (4)	DXA	LBM, FM, weight
Athithan et al., 2023a [81]	MGCS	810 kcal	12 weeks	15 (4)	T2DM	52.5 (5.1)	37.2 (6.8)	DXA	LBM, FM, weight
Athithan et al., 2023b [81]	MGCS	810 kcal	12 weeks	8 (4)	T2DM	46.6 (7.0)	36.0 (4.9)	DXA	LBM, FM, weight
Aukan et al., 2023 [82]	MGCS	750 kcal	10 weeks	15 (10)	Mixed (majority NDM)	45.5 (2.6) *	39.7 (0.9) *	ADP	FFM, FM, weight
Martins et al., 2023 [83]	SGCS	550–660 kcal	8 weeks	70 (41)	NDM	44 (9)	36.3 (4.0)	ADP	FFM, FM, weight
Näätänen et al., 2023a [84]	SGCS	600 kcal	7 weeks	51 (36)	NDM	50.5 (9.2)	34.0 (2.4)	BIA	FFM, FM, weight
Näätänen et al., 2023b [84]	SGCS	600 kcal	7 weeks	28 (22)	NDM	48.3 (8.6)	34.6 (2.8)	BIA	FFM, FM, weight

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
