# Peer review of "A Systematic Review and Meta-Analysis of the Effect of Caloric Restriction on Skeletal Muscle Mass in Individuals with, and without, Type 2 Diabetes"

_nutrients, 2024, doi:10.3390/nu16193328_

Round 1

Reviewer 1 Report

Comments and Suggestions for Authors

This systematic review and meta-analysis provided most comprehensive view on the effect of caloric restriction on skeletal muscle mass in individuals with and without type 2 diabetes. The rationale the results were well presented, and the implications of the findings appropriately discussed. It is commendable that the authors considered the relevance of muscle quality in the discussion. There is emerging evidence that it is the muscle quality (PMID: 36738835), but not the quantity (PMID: 35662615), that is closely linked to glycaemic outcomes in individuals with, or at risk of, type 2 diabetes. This issue may be commented further in the Discussion.

The manuscript is very well written. The rationale for the need of systemic review was clear and the methods were robust. Data were appropriated presented and interpreted. My suggestions to incorporate two additional points are intended to further strengthen the arguments, for which I suggest an opportunity of minor revision by the authors.

Author Response

Dear colleague,

Many thanks for the positive comments and the insight into this recent work regarding the importance of muscle strength. We believe this does enhance the arguments and have referred to these studies in the discussion - they have been included in lines 400-401.

Reviewer 2 Report

Comments and Suggestions for Authors

This is a thoughtfully, meticulously, and carefully performed meta-analysis on muscle mass and fat mass loss during low-calorie dieting.  I found no issues worth noting in the manuscript.

Author Response

Dear colleague, we are extremely grateful for your positive comments.

Reviewer 3 Report

Comments and Suggestions for Authors

The manuscript is a valuable review focused on the analysis of the effect of caloric restriction on the loss of muscle- and fat mass in individuals with, or without, T2DM. Carefully selected studies have provided convincing information showing that even less restrictive caloric interventions can lead to a reduction in body weight and fat mass, unfortunately also associated with muscle mass loss.

The topic of the article is very interesting and the study is well conducted. The manuscript is well written, recent findings are carefully presented and the results are well documented.

I have only minor comments:

1) The acronym VMD appears for the first time on line 182 and should be explained.

2) In Table 1, for the study: "van Dale, Schrijver & Saris 1990 [39]" the Age mean is 97.8. Is this not a mistake?

3) Since this is a scientific study, I recommend modifying the abbreviation T2D with the correct designation, i.e. T2DM.

4) The same applies to labeling body weight. Only the word "weight" is used throughout the manuscript.  „Weight“ should be edited as "Body weight" .

Author Response

Dear colleague,

Many thanks for the feedback, we are exrtremely grateful. Here are our responses:

Comment 1: We could not find a VMD acronym in line 182. This may have been something that edited incorrectly in the original manuscript but this is not present in the revised manuscript.

Comment 2: Thanks very much for highlighting this! It was an error and has been rectified. We have reviewed the entire table again to ensure all other data presented are accurate.

Comment 3: Thank you again for noting this. We have changed the abbreviation to T2DM throughout the manuscript and supplementary material.

Comment 4: Again, thank you for making us aware of this this. We have made the recommended changes and agree that the manuscript reads better now for it.

Kindest regards.